# Enhanced Water Solubility and Oral Bioavailability of Paclitaxel Crystal Powders through an Innovative Antisolvent Precipitation Process: Antisolvent Crystallization Using Ionic Liquids as Solvent

**DOI:** 10.3390/pharmaceutics12111008

**Published:** 2020-10-22

**Authors:** Qilei Yang, Chang Zu, Wengang Li, Weiwei Wu, Yunlong Ge, Lingling Wang, Li Wang, Yong Li, Xiuhua Zhao

**Affiliations:** 1College of Chemistry, Chemical Engineering and Resource Utilization, Northeast Forestry University, Harbin 150040, China; yangqilei1995@163.com (Q.Y.); carrie416@163.com (C.Z.); syngelzd@163.com (W.L.); wuweiwei0522@163.com (W.W.); wo-geyunlong@163.com (Y.G.); llwang@nefu.edu.cn (L.W.); kobe4813765@163.com (L.W.); liyong_dlpu@163.com (Y.L.); 2Key Laboratory of Forest Plant Ecology, Northeast Forestry University, Ministry of Education, Harbin 150040, China; 3Engineering Research Center of Forest Bio-preparation, Ministry of Education, Northeast Forestry University, Harbin 150040, China

**Keywords:** paclitaxel, antisolvent precipitation process, ionic liquids, bioavailability

## Abstract

Paclitaxel (PTX) is a poor water-soluble antineoplastic drug with significant antitumor activity. However, its low bioavailability is a major obstacle for its biomedical applications. Thus, this experiment is designed to prepare PTX crystal powders through an antisolvent precipitation process using 1-hexyl-3-methylimidazolium bromide (HMImBr) as solvent and water as an antisolvent. The factors influencing saturation solubility of PTX crystal powders in water in water were optimized using a single-factor design. The optimum conditions for the antisolvent precipitation process were as follows: 50 mg/mL concentration of the PTX solution, 25 °C temperature, and 1:7 solvent-to-antisolvent ratio. The PTX crystal powders were characterized via scanning electron microscopy, Fourier transform infrared spectroscopy, high-performance liquid chromatography–mass spectrometry, X-ray diffraction, differential scanning calorimetry, thermogravimetric analysis, Raman spectroscopy, solid-state nuclear magnetic resonance, and dissolution and oral bioavailability studies. Results showed that the chemical structure of PTX crystal powders were unchanged; however, precipitation of the crystalline structure changed. The dissolution test showed that the dissolution rate and solubility of PTX crystal powders were nearly 3.21-folds higher compared to raw PTX in water, and 1.27 times higher in artificial gastric juice. Meanwhile, the bioavailability of PTX crystal increased 10.88 times than raw PTX. These results suggested that PTX crystal powders might have potential value to become a new oral PTX formulation with high bioavailability.

## 1. Introduction

Polymorphisms refer to the variety of crystal structures created by different interactions between the same molecules in a solid state, which decides the physiochemical properties of crystals. Polymorph design of active pharmaceutical ingredients is normally regarded as one of the most important factors in drug development and production, as it determines the bioavailability of the drug and its stability [1]. Determination of distinct crystal habits is significantly influenced by solvent recrystallization because the morphology depends on the solute–solvent interaction on various crystal faces [2,3,4]. Drug crystallization methods include solution crystallization from single or mixed solvents [5,6,7], supercritical fluids crystallization [8], seeding strategies [9], capillary crystallization [10], polymer-induced heteronucleation [11], heteronucleation on substrates [12], and laser-induced nucleation [13]. Solution crystallization is one of the most popular methods in drug crystallization.

Crystallization conditions include solvent/antisolvent species, temperature, additive, supersaturation, agitation etc. In these conditions, solvent/antisolvent species is often considered the most important factor that determines the polymorphs in crystallization.

Ionic liquids (ILs), which are also known as room-temperature ionic liquids (RTILs), are salts that are composed only of ions and have a melting point below 100 °C [14]. RTILs are used as “green” solvents to replace traditional volatile organic solvents, and it is used in a number of research fields, due to their low melting points, such as synthetic chemistry, catalysis, separation technologies, and bio-enzymatic reactions. In particular, recent research progressed to use them as media for carrying out crystallization. A number of publications emerged for their use in crystal engineering applications for inorganic molecules. [15] For example, polymorphic paracetamol can be successfully crystallized from 1-hexyl-3-methylimidazolium hexafluorophosphate and 1-Butyl-3-methylimidazolium hexafluorophosphate, using cooling crystallization [16]. Among them, the 1-hexyl-3-methylimidazolium bromide (HMImBr) is widely used in the extraction of bioactive natural products because of its advantages, such as selective extraction ability, unique advantages of ideal thermal stability, comprehensive dissolving capacity, negligible vapor pressure, excellent structural designability, etc. However, there was no report on the HMImBr as a medium for drug crystallization [17,18,19,20]. Moreover, the anti-solvent crystallization technique using ILs as a solvent is one of the new methods to improve the crystal structure of drugs [21]. In this emerging area, there are relatively few publications that have explored this avenue of research. According to An’s research [22], they used ILs as both solvent and antisolvent for designing crystallization, to modify the crystal morphology and crystal structure of adefovir dipivoxil. However, ILs as antisolvents are not good cleaning agents, so the prepared drug cannot be used clinically. The methods using ILs as solvent and water as antisolvent to prepare new crystal drugs could become greener, cheaper, and easier to scale and industrialize.

Paclitaxel (PTX) is a diterpenoid anticancer agent that was discovered in the bark of the yew tree [23]; the structure of PTX is shown in Figure 1a. PTX is one of the most effective anticancer drugs for treating ovarian cancer, breast cancer, Kaposi’s sarcoma, and non-small-cell lung cancer [24,25]. Its application was also expanded to the treatment of acute rheumatoid arthritis [26,27] and Alzheimer’s disease [28]. However, PTX, which has the brand name Taxol^®^, is administered in combination with Cremophor EL as an adjuvant, because of the poor solubility of PTX in aqueous media. Unfortunately, Cremophor EL induces hypersensitivity and causes serious side effects in many patients [29]; therefore, finding a new formulation with high water solubility and low cost is necessary. Oral administration is the easiest and most convenient route of drug delivery. However, PTX oral bioavailability is very low mainly because of its poor aqueous solubility. To solve this problem, several approaches were reported, such as solubilization with d-α-tocopherol polyethylene glycol 1000 succinate (TPGS) 400 [30], co-administration with a P-glycoprotein inhibitor [31], microemulsions or nanoemulsions [32], formation of inclusion complexes with cyclodextrins, and entrapment in nanoparticles and polymer micelles [33,34]. However, these methods all have different stability and toxicity problems. Preparation of new PTX crystals powders might be a better method to enhance solubility and oral bioavailability. Designing PTX crystallization to enhance solubility with antisolvent crystallization technique using ILs is yet to be reported.

This research aims to prepare PTX crystal powders with high water solubility and bioavailability through an antisolvent crystallization technique using 1-hexyl-3-methylimidazolium bromide (HMImBr) (structure is shown in Figure 1b) as solvent and water as antisolvent. Single-factor design was utilized to obtain the optimal preparation conditions. Furthermore, the morphology, crystalline state and chemical structure, dissolution rate, and bioavailability of PTX crystal powders were investigated.

## 2. Materials and Methods

### 2.1. Materials

PTX (99.2% purity) was purchased from Chongqing Taihao Pharmaceutical Co., Ltd. (Chongqing, China). HMImBr was purchased from Shanghai Cheng Jie Chemical Co., Ltd. (Shanghai, China). Analytical-grade ethanol, ethyl ether, and other reagents were obtained from Beijing Chemical Reagents Co. (Beijing, China). Acetonitrile, methanol, and tert-butyl methyl ether were HPLC grade. Deionized water was purified through a Milli-Q water purification system from Millipore (Bedford, MA, USA).

### 2.2. Preparation of PTX Crystal Powders

PTX crystal powders were prepared using the antisolvent precipitation technique. HMImBr was used as the solvent, and water was used as the antisolvent. PTX–HMImBr solution was then injected into a beaker with water, under stirring, at room temperature. The solid drug immediately precipitated upon mixing. After a certain time, the suspension was centrifuged until only a layer of precipitation was left. The suspension was then repeatedly washed with water for 8–10 times, through repeated centrifugation, and the particles were then freeze-dried. Every experiment was repeated at least thrice. The schematic diagram of the preparation of antisolvent process is shown in Figure 2.

### 2.3. Optimization of the Antisolvent Precipitation Process

Single-factor experiments were performed to determine the optimum conditions for the preparation of PTX crystal powders. Three factors were chosen for analysis in a preliminary experiment—drug concentration, antisolvent-to-solvent volume ratio, and temperature. In this method, only one factor was changed, whereas the others remained constant. When drug concentration was studied, the other factors were set under the following conditions—the antisolvent-to-solvent volume ratio was 5, and the temperature was 25 °C.

The same method as above was applied to study the other factors. The effects of drug concentrations at 10, 30, 50, and 70 mg/mL were studied. The antisolvent-to-solvent volume ratio was varied at 3:1, 5:1, 7:1, and 9:1. The temperature was set at 25, 50, 75, and 95 °C and investigated. Finally, the optimum condition for every factor was determined based on the water solubility of the obtained PTX crystal.

### 2.4. Characterization of the PTX Crystal Powders

#### 2.4.1. Scanning Electron Microscopy (SEM) and Mean Particle Size (MPS) Detection

The morphologies of raw PTX particles and PTX crystal powders were detected using SEM (FEI, Eindhoven, The Netherlands). The samples were prepared through direct deposition onto a carbon tape placed on the surface of an aluminum stub. Before analysis, the samples were coated with gold for 4 min, using a sputter coater (KYKY SBC-12, Beijing, China).

#### 2.4.2. Fourier Transform Infrared Spectroscopy (FT–IR)

The chemical components of the raw PTX and PTX crystal powders were analyzed through FT–IR spectroscopy using an IRAffinity-1 spectroscope (Shimadzu Corporation, Kyoto, Japan). Every sample was mixed with KBr, using an agate mortar and was compressed into a thin disc. The scanning range was 4000–400 cm^−1^ with a resolution of 4 cm^−1^.

#### 2.4.3. High-Performance Liquid Chromatography–Mass Spectrometry (HPLC–MS)

The raw PTX and PTX crystal powders, each of which was at 1 mg/mL concentration, were dissolved separately in methanol. The sample was diluted as necessary. The HPLC–MS spectra were obtained using the analyst1.4 of API3000 (API, Frederick, MD, USA). The mass spectrometer was operated in the negative ion mode.

#### 2.4.4. X-Ray Diffraction (XRD)

The crystal forms of the samples were also detected using X-ray (Bruker, Karlsruhe, Germany) with nonmonochromated CuKa radiation. The samples were scanned from 4° to 50° at rates of 10° per minute 0.02° per minute. The crystal forms were then measured at a voltage of 40 kV and 40 mA.

#### 2.4.5. Differential Scanning Calorimetry (DSC)

A total of 5 mg raw PTX and PTX crystal powders were investigated using DSC (TA instruments, NewCastle, DE, USA) at an increasing temperature heating rate of 10 °C/min from 50 °C to 300 °C.

#### 2.4.6. Thermogravimetric (TG) Analysis

The samples were analyzed using a TG analyzer (Diamond TG/DTA from Perkin–Elmer, Waltham, MA, USA). Approximately 5 mg samples were weighed in open aluminum pans. The test condition was as follows—nitrogen flow rate of 50 mL/min. The percentage weight loss of the samples was monitored from 30 °C to 600 °C with an increasing heating rate of 10 °C/min.

#### 2.4.7. Raman Spectroscopy

The Raman spectra of the specimens were collected using a computer-controlled laser Raman microprobe equipped with a Leica DM/LM optical microscope, with a 100× objective (NA, 0.9) and a charge-coupled device detector attached to a modular research spectrograph (Renishaw InVia, Renishaw PLC, Gloucestershire, UK). The 100× objective was used to increase the precision of the acquired chemical data, resulting in a laser spot size of ≤1 μm. A monochromatic, near-infrared diode laser operating at 633 nm was used to induce the Raman scattering effect. The spectral coverage of this model ranged from 100 cm^−1^ to 3450 cm^−1^ with an average spectral resolution of 5 cm^−1^. Before each experiment, a comparison with a silicon standard was performed using the calibration system integrated with the software (WiRE 2.0, Renishaw, Gloucestershire, UK), according to manufacturer’s specifications to calibrate the wavelength and intensity. The exposure time for each scan was 40 s, and the laser power on the specimen was 8 mW.

#### 2.4.8. Solid-State Nuclear Magnetic Resonance (SSNMR)

All measurements were performed using a broadband 3.2 mm solid-state nuclear magnetic resonance (ssNMR) probe and a 300 MHz Bruker BioSpin GmbH (Bruker, Karlsruhe, Germany) spectrometer. Samples were prepared by packing an adequate amount of each sample, as received, into 3.2 mm silicon nitride (Si_3_N_4_) SSNMR rotors. The sample spinning rate was set to 10 kHz. The 13C NMR spectra (100 scans) were recorded using the pulse sequence of cross-polarization magic angle spinning. Prior to each spectrum, the corresponding T1 constant (spin–lattice relaxation) was measured using the pulse sequence of saturation recovery.

### 2.5. Dissolution Study

#### 2.5.1. Preparation of the Tablets Containing the PTX Crystal

The raw PTX and PTX crystal powder complexes were compressed into tablets based on formulations. Microcrystalline cellulose (Avicel PH101, 50 mg/tablet) and Ac-Di-Sol (20 mg/tablet) were added as tablet diluent and disintegrant. The initial 3 mg and 12 mg weight of the PTX samples were converted to 200 mg by adjusting the amount of lactose accordingly. All ingredients of the tablet formulation were mixed for 15 min in a plough shear blender at 400 rev/min. The mixture was blended again for 1 min at 400 rev/min and then compressed into tablets with a total weight of 200 mg, using an Erweka EKO (Rieckermann, Hamburg, Germany) [35].

#### 2.5.2. Kinetic Solubility Test

A total of 3 mg tablet of raw PTX and PTX crystal powder were added into 1 L deionized water, and then 12 mg tablet of raw PTX and PTX crystal powder were added into the dissolution medium (artificial gastric juice with 5 mL Tween-80). The dissolution temperature and paddle speed were set at 37 ± 0.5 °C and 100 r/min. Consequently, 10 mL aliquots were withdrawn each time at 0.5, 1, 2, 3, 4, 6, 8, 12, 16, 24, 36, and 48 h, and filtered using a 0.22 mm filter. Each 9 mL filtrate of the samples were mixed with 1 mL chloroform, vortexed for 1 min, and centrifuged for 10 min at 5000 rpm. Subsequently, 800 µL of the organic layer was transferred to a clean vial and blow dried at 40 °C, under a gentle stream of nitrogen. The dried residue was then reconstituted in 200 µL methanol. After being vortexed for 2 min, the content was centrifuged at 12,000 rpm for 10 min. In addition, 20 µL supernatant was injected for HPLC analysis.

Chromatographic analyses were performed on a Waters HPLC system consisting of a pump (model 1525), an autosampler (model 717 plus), and a UV detector (2487 dual λ absorbance detector). The C18(2)column (Diamonsil, 5 μm, 4.6 mm × 250 mm, Dikma Technologies, Beijing, China) was used at 30 °C. Methanol, acetonitrile, and water (52.2:22.5:25) was used as the mobile phase. The flow rate was 1.0 mL/min, and the injection volume was 20 µL. The signal was monitored at 227 nm. The linear regression equations for the reference compound of PTX was *Y*ptx = 3.6 × 104*X* − 24656 (r^2^ = 0.9998). A good linearity was found for PTX in the range of 0.3–300 µg/mL.

### 2.6. Bioavailability Study of Paclitaxel Crystal Powders in Rats

SD rats (230 ± 10 g) provided by the Harbin Medical University (Harbin, Heilongjiang, China), were used in the in vivo experiments. The rats were deprived of food overnight before the experiment. The animal use and care protocol was reviewed and approved by the ethics committee of Harbin Medical University. Two groups of the rats (*n* = 6) were orally administered with 10 mg/kg of PTX through gavage, using an oral feeding sonde, under ether anesthesia. The PTX dose given to each rat was calculated based on the weight of the rat, at a dose of 10 mg/kg. The blood samples were taken from the eye-ground venous plexus at predetermined time points after 0.5, 1, 1.5, 2, 3, 4, 6, 8, 10, 12, 24, and 48 h. The blood samples were placed on ice (4 °C) and centrifuged for 10 min at 3000 rpm, and plasma aliquots were stored at −40 °C, until additional extraction and analysis.

Frozen samples were thawed at room temperature and treated as follows. An aliquot (100 μL) of plasma sample was mixed with 25 μL of the internal standard solution (docetaxel, 4 μg/mL in methanol, previously evaporated). After vortex mixing, the liquid–liquid extraction was accomplished with the addition of 4 mL of tert-butyl methyl ether, following gentle vortex agitation (1 min). The mixture was centrifuged for 10 min at 5000 rpm, and then the organic layer was transferred to a clean vial and blow dried at 40 °C, under a gentle stream of nitrogen. Finally, the residue was dissolved in 100 μL reconstitution solution (0.01 M acetonitrile–phosphate buffer, pH = 2, 50/50 *v*/*v*), transferred to autosampler vials, capped, and placed in the HPLC autosampler. A total of 20 μL aliquot of each sample was injected onto the HPLC column [32].

### 2.7. Statistical Analysis

All experiments were conducted in triplicates and repeated at least three different times. Differences between the groups were assessed by one-way ANOVA, and a *p* value less than 0.05 was considered to be significant.

## 3. Results and Discussion

### 3.1. Optimization of Antisolvent Precipitation Process

The factors influencing the saturation water solubility of PTX crystal powders were optimized using single-factor design. Three factors were chosen for the analysis—drug concentration, antisolvent-to-solvent volume ratio, and temperature.

As shown in Figure 3a, the saturation solubility in water of the PTX crystal powders increased slowly under 30 mg/mL concentration and increased sharply in the range of 30–50 mg/mL concentration. However, when the concentration reached 50 mg/mL, the saturation solubility in water of PTX crystal powders decreased dramatically. Therefore, 50 mg/mL was considered as the optimal concentration of the PTX solution for the antisolvent precipitation process. As shown in Figure 3b, the saturation water solubility of the PTX crystal powders decreased with the increase of temperature. Thus, 25 °C was considered as the optimal temperature for the antisolvent precipitation process. Figure 3c indicates that the saturation solubility in water of PTX crystal powders initially decreased slowly and then increased sharply when the solvent-to-antisolvent volume ratio increased. However, when the solvent-to-antisolvent volume ratio was 1:7, the water saturation solubility of the PTX crystal powders decreased dramatically. Therefore, 1:7 was considered as the optimal solvent-to-antisolvent volume ratio for the antisolvent precipitation process.

Using the single-factor design, the optimum antisolvent precipitation process conditions for PTX were found to be as follows—50 mg/mL concentration of the PTX−HMImBr solution, 7:1 antisolvent-to-solvent volume ratio, and 25 °C temperature. Based on the confirmatory water saturation solubility tests, better PTX crystal powders were obtained with water saturation solubility of 6.052 µg/mL. The solubility in water of the PTX raw material and PTX crystal powder was 1.16 µg/mL and 6.05 µg/mL. PTX crystal powders had an increased solubility of nearly six times in the water. Thus, it feasible to prepare PTX crystal through the antisolvent precipitation process, using HMImBr as a solvent for improving the water solubility of PTX.

### 3.2. Morphology

Figure 4a,b show the SEM of raw PTX particles and PTX crystal powders that were made using the antisolvent precipitation process. SEM analysis showed that raw PTX particles had cuboid structure crystals (Figure 4a), and most of the crystal powders had uniform short rod cuboid structure (Figure 4b). The particles of the raw PTX were big and had a varied distribution, whereas PTX crystal powders were small in size and had a uniform distribution.

### 3.3. Chemical Structure

The FT–IR spectra of raw PTX and PTX crystal powders are compared in Figure 5. The spectrum showed that absorption peaks of the two have the same shape and frequency position in the 400–4000 cm^−1^ range, indicating that the functional group structures of PTX raw material and crystal powders were not significantly different.

The chemical structures of the two PTX samples were further evaluated using HPLC–MS to determine their molecular weights. As shown in Figure 6, the molecular weights were not altered. Based on the calculation of the mass spectrum, the molecular weight of raw PTX and the PTX crystal powder was 854.3 and 854.4. Combining the FT–IR and HPLC–MS results, we could determine that the PTX chemical structure did not change during the antisolvent recrystallization process.

### 3.4. The Crystal Structure

The raw PTX and PTX crystal powders were subjected to XRD analysis to obtain information on crystalline change after the antisolvent recrystallization process. Figure 7a and Table 1 show the XRD results for the raw PTX. As illustrated, the two samples had significantly different diffractograms. The raw PTX showed several characteristic peaks at 2θ = 5.498°, 8.866°, 10.007°, 11.171°, 12.368°, 13.897°, 14.506°, 15.596°, 17.017°, 18.727°, and 21.933°, whereas the PTX crystal powders (Figure 8b) exhibited several characteristic peaks at 5.068°, 5.923°, 9.562°, 10.794°, 13.491°, 13.667°15.225°, 16.312°, and 20.146°. This finding suggests that the PTX particles changed their crystal structures after the antisolvent processing.

Raman spectra (Figure 8) show the raw PTX in the range of the C–H stretching vibrations of the alkyl (2860–3200 cm^−1^), the valence vibrations of the carbonyl and amine group (1900–1500 cm^−1^), and the C–H stretching vibrations of the benzene ring from 900 cm^−1^ to 650 cm^−1^. However, the most striking differences between the raw PTX and PTX crystal powders was that the PTX crystal powders exhibited no significant stretching and vibration. These differences in Raman spectra were attributed to the differences of intramolecular hydrogen bonds and intermolecular Van der Waals force; these differences could also indicate the change of crystalline structure of PTX after the antisolvent recrystallization processing.

SSNMR was used to investigate the chemical structure, because it is a sensitive indicator of hydrogen bonding [36,37,38,39], stereochemistry [40], conformation [41], steric forces [42], and electrostatic interactions [43,44,45]. The PTX crystal structure (Figure 1) contained two molecules per asymmetric unit (i.e., prime= 2 referred to herein as molecules 1a and 1b) and crystallizes in the P21 space group [46]. The two molecules of PTX were 10-deacetylbaccatin III [47] and phenylisoserine side chain [48]. The SSNMR analysis of these data provided shift assignments to all molecular positions (Table 2). The different chemical shifts could be found at C16 and C17, by comparing the chemical shift principal values (CSPV) of raw PTX with the PTX crystal. This phenomenon was attributed to the increased bond angle of C16–C17 than raw PTX, in 10-deacetylbaccatin III (Figure 9). The bond angle of C16–C17 changed because the space structure of the unit cell was loose, and the loose structure allowed easier collection of the water molecule. Thus, the PTX crystal powder had a better water solubility.

The DSC is a technique used to measure the temperature and energy variations involved in the phase transitions; these variations are related to the degree of crystallinity and stability of the solid state of drug. From Figure 10, the DSC thermogram of the PTX revealed one endothermic peak at 221.99 °C and one exothermic peak at 244.6 °C, which were attributed to the melting and decomposition of PTX, respectively. While the endothermic peak and the exothermic peak the PTX crystal powders were at about 147.88 and 211.7 °C, respectively. This phenomenon was in agreement with the Raman and XRD analysis, which showed that the raw PTX crystal structure had changed, as confirmed by the decreased melting point of raw PTX. The decreased melting point indicated that the band energy also decreased, which coincided with the bond angle change of C16–C17.

As shown by the TG curve (Figure 11), when the temperature increased from 40 °C to 240 °C, the raw PTX had nearly no weight loss, and PTX weight decreased quickly at a temperature of approximately 240 °C. PTX crystal powders began to slowly lose weight from 40 °C to 220 °C and immediately lost weight at approximately 220 °C. The total weight loss was approximately 18.83% and 22.15% for the raw PTX and PTX crystal powders. These results showed that both kinds of crystals had good thermal stability.

### 3.5. Dissolution Results

HPLC analysis results showed that at concentrations of 1.16 µg/mL and 6.05 µg/mL, the raw PTX and PTX crystal powder were soluble in pure water. Moreover, at a concentration of 10.42 µg/mL and 24.65 µg/mL, the raw PTX and PTX crystal powder were soluble in artificial gastric juice. The solubility of PTX crystal powders was increased nearly six times in water and 2.4 times in artificial gastric juice. Thus, it is feasible to prepare PTX crystal powders through the antisolvent precipitation process to improve and enhance its solubility in water.

The dissolution profiles of the raw PTX and PTX crystal powders in water are shown in Figure 12a. The dissolution rate of the PTX crystal powder increased rapidly from 25.09% to 50.67% within 8 h, and then its dissolution ceased as the time increased. By contrast, only a small amount of the drug was dissolved from the raw PTX at a steady rate within 6 h, and the maximum dissolution rate of the raw PTX was 15.80%. The solubility test in artificial gastric juice for each PTX sample was performed over 48 h. As shown in Figure 12b, the dissolution rate of the PTX crystal powders increased slowly from 5.56% to 32.50% and that of the raw PTX increased from 2.36% to 25.56% within 36 h. Consequently, both of them no longer exhibited dissolution as the time increased. The dissolution results proved that the bond angle of C16–C17 changed and that the crystal structure increased the dissolution rate.

### 3.6. Bioavailability Study

Figure 13 shows the pharmacokinetic profiles of each sample. The PTX concentration in rat plasma of the PTX crystal powders and the raw PTX group reached the maximum of 15.71 µg/mL and 0.50 µg/mL after 2 h and 1 h of drug administration. The PTX crystal powders dramatically decreased from 15.71 µg/mL to 1.90 µg/mL, within 2 h to 6 h, and slowly increased to 8.30 µg/mL within 6 h to 10 h. Then, the PTX concentration decreased until the concentration was nearly 0 µg/mL. The results showed that the PTX crystal powders had a second peak. This might be due to the liver and intestine circulation or gastric emptying in specific intervals, which enabled the drug to reach the small intestine twice. This resulted in two entries into the blood, which led to the appearance of double peaks. The same trends were exhibited by raw PTX; the raw PTX significantly decreased from 0.50 µg/mL to 0.21 µg/mL within 1 h to 4 h, and slowly increased to 0.38 µg/mL within 12 h to 16 h. Then, the PTX concentration decreased until the concentration was nearly 0 µg/mL. The bioavailability of the PTX crystal increased 10.88 times than raw PTX. The significant enhancement of oral bioavailability was also in agreement with the result of the dissolution test. In addition, previous research showed that the bioavailability of the super-antiresistant PTX micelles by a one-step method was almost twice that of the raw PTX [49]. However, the bioavailability of the PTX crystal powders prepared in this study were approximately 10.88-fold greater than that of raw PTX, which illustrated that the PTX crystal powders obtained in this study was remarkably effective in improving the extent of absorption of the PTX, and exhibited potential as a new oral drug formulation for clinical applications.

## 4. Conclusions

In this study, PTX crystal powders were prepared through the antisolvent precipitation process, using HMImBr as solvent and water as antisolvent. The factors influencing the water saturation solubility of the PTX crystal powders were optimized through a single-factor design, and the optimum conditions were determined as follows—50 mg/mL concentration of PTX–HMImBr solution, 1:7 solvent-to-antisolvent volume ratio, and 25 °C reaction temperature. Under the above conditions, the PTX crystal powder were obtained with the saturation solubility of 6.05 µg/mL, which was six times higher than that of the raw PTX. The PTX crystal powders obtained were characterized using SEM, FT–IR, HPLC–MS, Raman spectroscopy, XRD, DSC, SSNMR, and TG analysis. The SEM showed that PTX crystal powders were small in size and had a uniform distribution and short rods. Furthermore, the FT–IR and LC–MS results showed that the crystal powders had the same chemical structure as the raw drug. Meanwhile, the XRD, SSNMR, Raman spectra, and DSC analyses proved the change of the crystal structure. The XRD results indicated that the crystal powders had different peaks from that of the raw PTX and that CSPV also varied between the raw PTX and PTX crystal powders at C16 and C17 in 10-deacetylbaccatin III, showing that the bond angle of C16–C17 changed. The DSC curve showed that the melting point of the PTX crystal powders was 147.88 °C; the smaller melting point proved the lower bond energy, and this conclusion was consistent with the altered bond angle of C16–C17. The TG analysis showed that the raw PTX and PTX crystal powders had similar thermal stability. Furthermore, the maximum solubility of the PTX crystal powder water in water was 5.6 times higher than the raw drug. The dissolution rate and solubility of PTX crystal powders was 3.21 times higher compared to that of the raw PTX in water and was 1.27 times higher than the raw PTX in artificial gastric juice. The oral bioavailability of the PTX crystal increased 10.88 times than raw PTX. This antisolvent precipitation technique is a promising and economical method for the preparation of highly water-soluble drug particles for commercial use. Moreover, this method provides a good theoretical and practical basis for the development of other fat-soluble drugs.

## Figures and Tables

**Figure 1 pharmaceutics-12-01008-f001:**
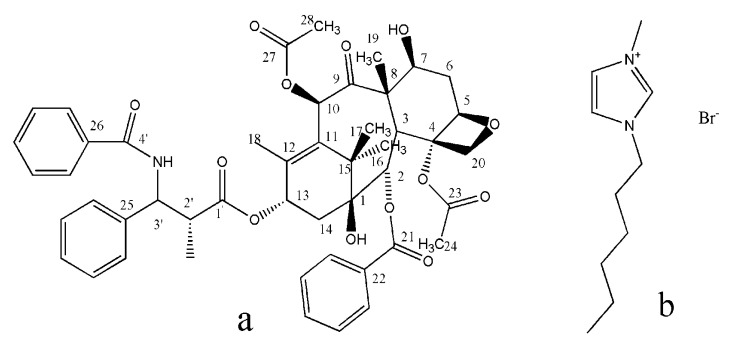
(**a**) Chemical structure of the paclitaxel. (**b**) Chemical structure of the HMImBr.

**Figure 2 pharmaceutics-12-01008-f002:**
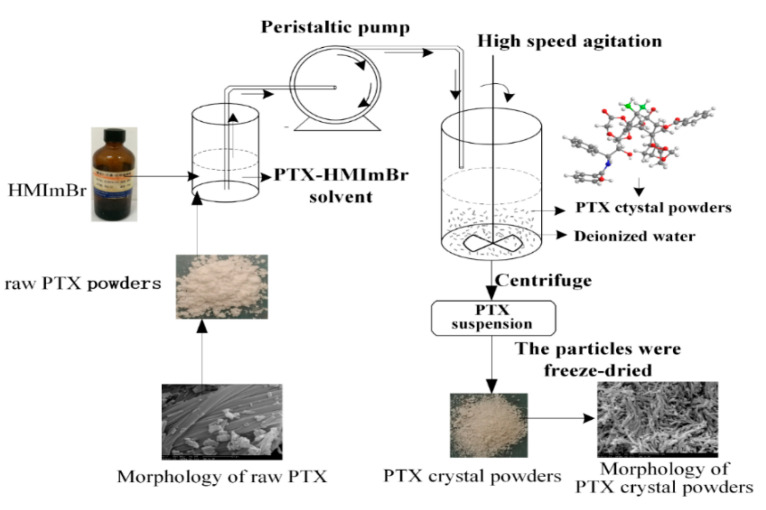
Schematic diagram of the experimental processes to prepare the paclitaxel (PTX) crystal powders.

**Figure 3 pharmaceutics-12-01008-f003:**
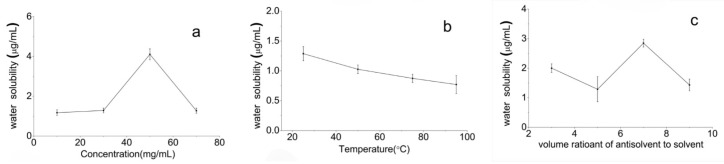
The effect of each parameter on the water saturation solubility of the PTX crystal powder. (**a**) Drug concentration; (**b**) temperature, and (**c**) volume ratio of antisolvent-to-solvent.

**Figure 4 pharmaceutics-12-01008-f004:**
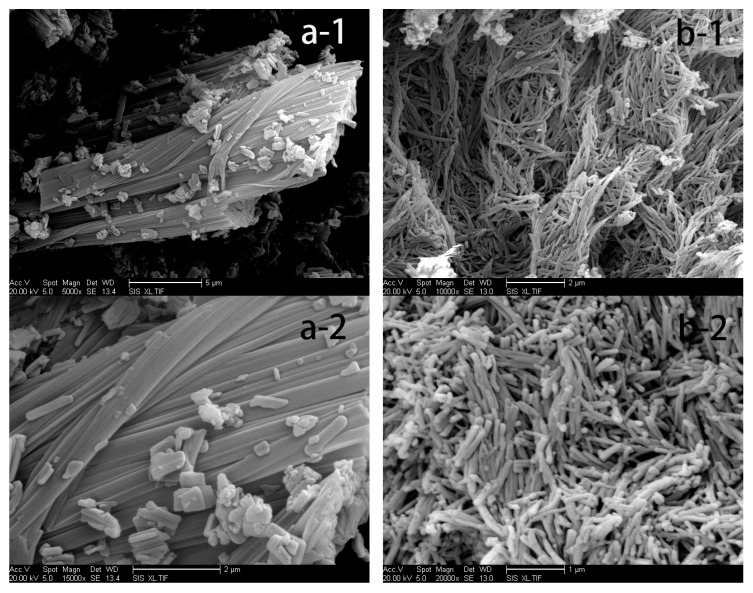
Scanning electron micrographs of each sample. (**a-1**) Raw PTX (magnification of 5000); (**a-2**) Raw PTX (magnification of 15,000); (**b-1**) PTX crystal powder (magnification of 10,000) and (**b-2**) PTX crystal powder (magnification of 20,000).

**Figure 5 pharmaceutics-12-01008-f005:**
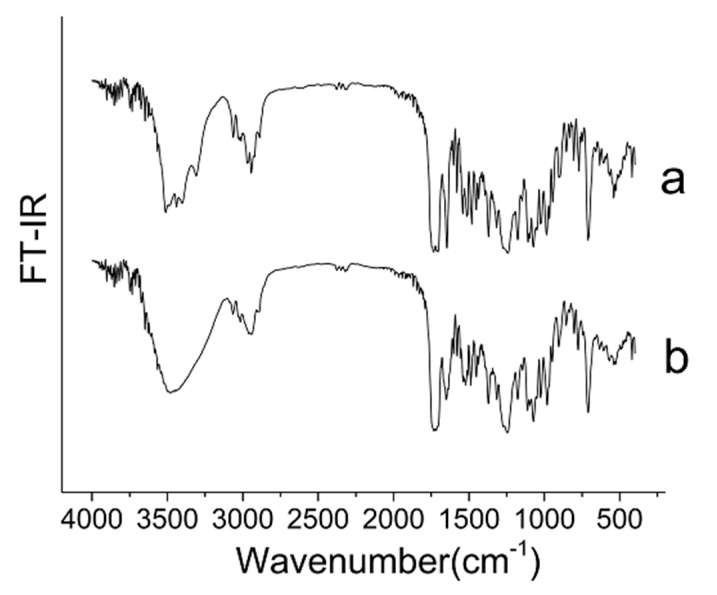
FT–IR spectra (Wavenumber: 400–4000 cm^−1^) of the samples. (**a**) Raw PTX; and (**b**) PTX crystal powders.

**Figure 6 pharmaceutics-12-01008-f006:**
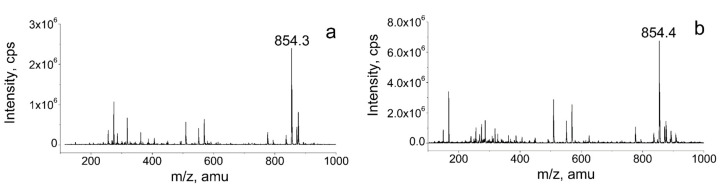
LC–MS spectra of the samples. (**a**) Raw PTX; and (**b**) PTX crystal powders.

**Figure 7 pharmaceutics-12-01008-f007:**
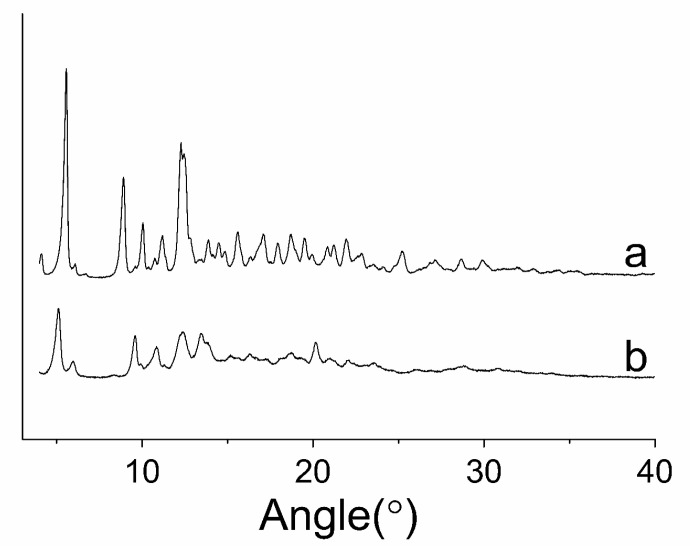
XRD patterns (2θ = 4–50°) of the samples. (**a**) Raw PTX; and (**b**) PTX crystal powders.

**Figure 8 pharmaceutics-12-01008-f008:**
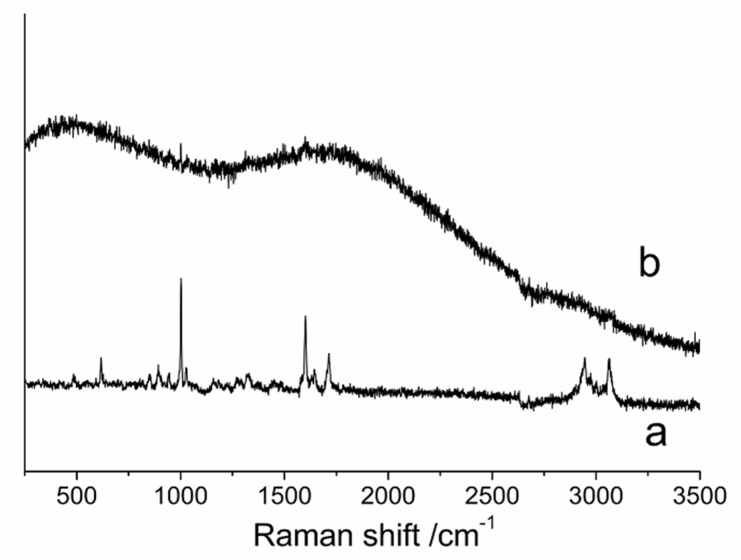
Raman spectra of the samples. (**a**) Raw PTX, and (**b**) PTX crystal powders.

**Figure 9 pharmaceutics-12-01008-f009:**
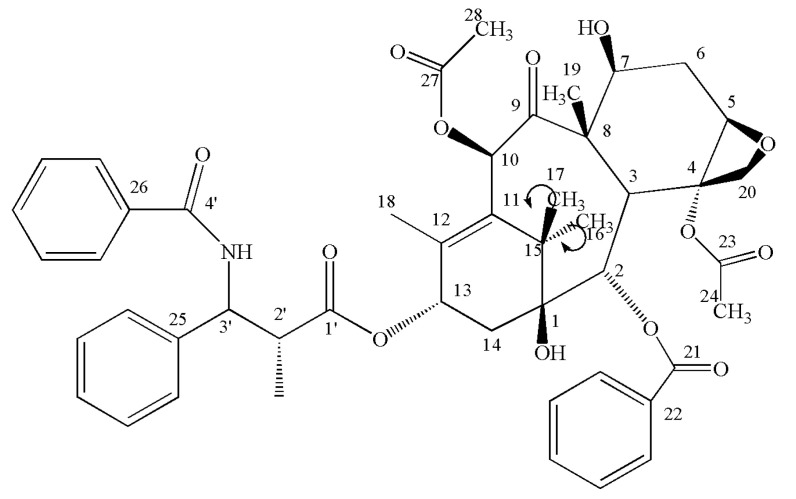
The change of crystalline structure of the PTX.

**Figure 10 pharmaceutics-12-01008-f010:**
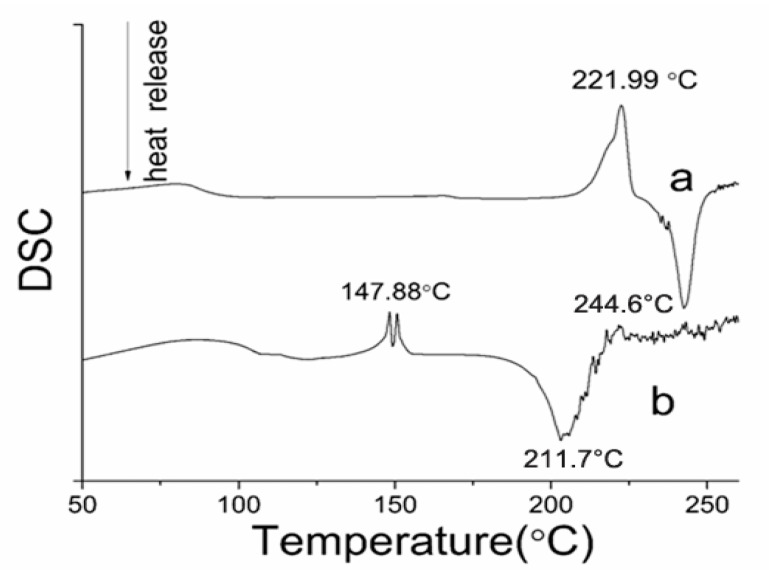
DSC curves (Temperature range: 50–3000 °C) of the samples. (**a**) Raw PTX, and (**b**) PTX crystal powders.

**Figure 11 pharmaceutics-12-01008-f011:**
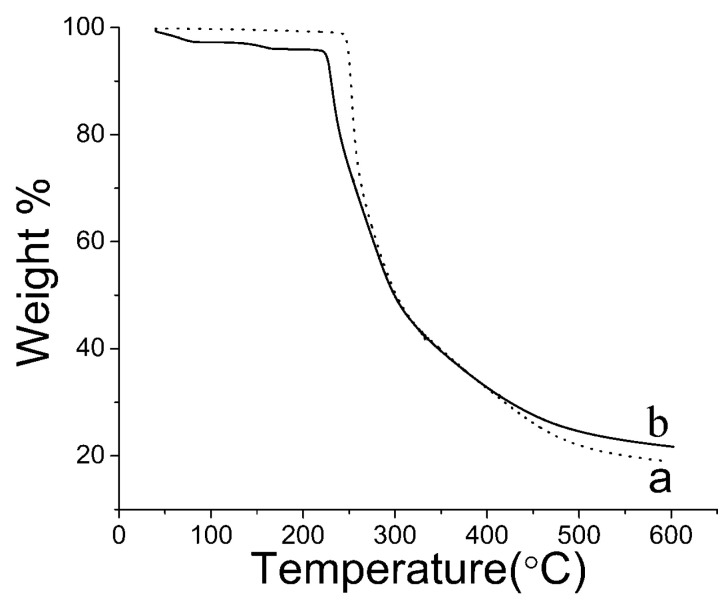
TG curve (Temperature range: 30–600 °C) of the samples. (**a**) Raw PTX; and (**b**) PTX crystal powders.

**Figure 12 pharmaceutics-12-01008-f012:**
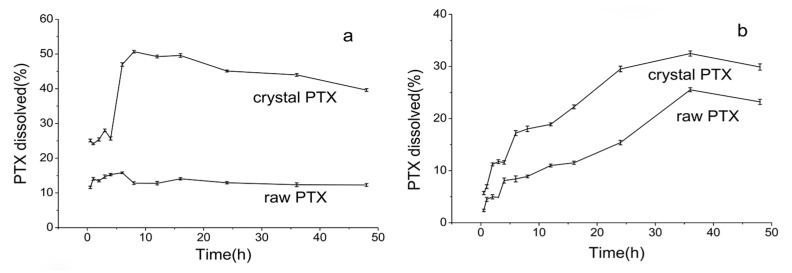
The dissolution profiles of the samples in water (**a**), and in artificial gastric juice (**b**).

**Figure 13 pharmaceutics-12-01008-f013:**
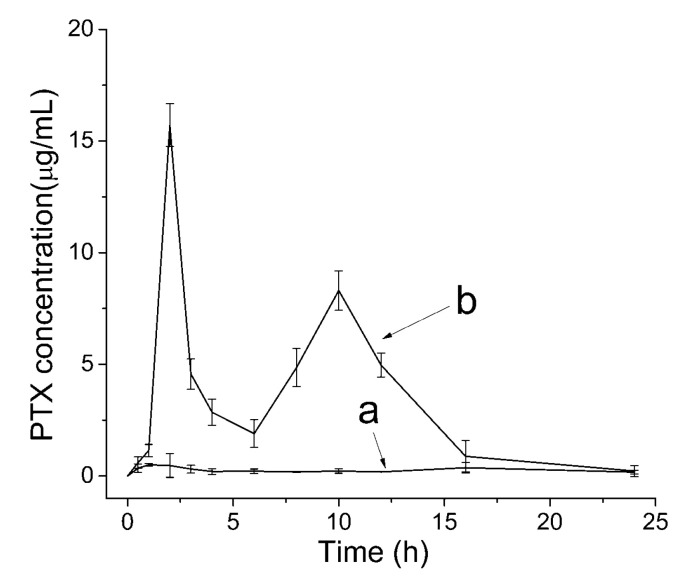
Mean (±SD) plasma concentration-time curves after oral administration of raw PTX (**a**) and PTX crystal powders suspension (**b**) at a dose of 10 mg/kg for PTX (*n* = 6).

**Table 1 pharmaceutics-12-01008-t001:** Analysis date of XRD.

Angle(°)	
Raw PTX	5.58°, 6.12°, 8.89°, 9.67°, 10.04°, 11.06°, 12.29°, 13.83°, 15.61°, 17.07°, 18.68°, 19.56°, 21.89°
Crystal PTX	5.19°, 6.12°, 9.82°, 11.13°, 12.50°, 13.59°, 16.49°, 20.34°

**Table 2 pharmaceutics-12-01008-t002:** Partial 13C shift assignments in solid paclitaxel.

Position	δ ^13^C/ppm (Crystal PTX)	δ ^13^C/ppm (Raw PXT)
C18	6.10	5.79
C16, C17, C19	16.35	15.50
C24, C28	28.99, 31.18	30.14
C6	37.58	37.19, 41.29
C8, C3’	51.53	51.62
C7, C13, C2’	68.64	68.93,
C4, C5	86.75	88.62
C22, C25, C26	123.18	123.10
C11, C12	134.56	134.16, 136.04
C21, C23, C4’	164.97	165.10

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
