# Peer review of "Enhanced Water Solubility and Oral Bioavailability of Paclitaxel Crystal Powders through an Innovative Antisolvent Precipitation Process: Antisolvent Crystallization Using Ionic Liquids as Solvent"

_pharmaceutics, 2020, doi:10.3390/pharmaceutics12111008_

Round 1
Reviewer 1 Report
It is an interesting and well-written paper, that brings useful results, together with innovative approach.
I suggest the authors to address the following issues before acceptance:
1) the use of an ionic liquid is very interesting, but may result in toxic residue - please, comment about this topic.
2) mean size and size distribution of the crystals produced should be incorporated to the paper.
3) I think in vitro and/or in vivo (zebra fish, e.g.) would greatly improve the quality and atractiveness of the paper - please, consider such possibility.
Author Response
Response to the Reviewers
Replies to the Reviewers
First of all, we would like to thank the reviewers for their positive and constructive comments and suggestions.
Replies to Peer Reviewer: 1
Reviewers' comments:
(1) The use of an ionic liquid is very interesting, but may result in toxic residue - please, comment about this topic.
Answer: Thank you very much for your suggestion. It is difficult to detect the residual problem of the ionic liquid in our laboratory, but the HMImBr, as an ionic liquid, is a kind of green solvent with low toxicity, and the obtained PTX crystal powder has been washed with water for 8-10 times in the preparation process, so the residual amount of the HMImBr is very low.
(2) Mean size and size distribution of the crystals produced should be incorporated to the paper.
Answer: Thank you very much for your suggestion. Because the PTX crystal powder is crystalline, not spherical, so there is no way to determine its particle size.
(3) I think in vitro and/or in vivo (zebra fish, e.g.) would greatly improve the quality and atractiveness of the paper - please, consider such possibility.
Answer: Thank you very much for your suggestion. I think your opinion is very good, and we will do some related research in the future.
Reviewer 2 Report
Overall, the manuscript is well-written nd of interest to the readers of Pharmaceutics. However, there are few issues that should be addressed before publication:
- Have you considered to use Design of experiments instead of the OFTA (One factor at a time) methodology? I think this will increase the optimisation step of your formulations.
- Why did you start the DSC at 50 degrees?
- The FTIR and Raman figures are not visible.
- You are claiming that you have a new polymorph of PTX base don the different Bragg peaks. Then, it is required to check the crystal structure of the new polymorph and uploaded into the Cambridge data base.
- What is the peak at 221 degrees of your DSC? Are you sure the PTX does not degrade before melting?
- You should explain why you are getting your second peak in the PK study. I see two main reasons one that your drug is absorbed by lymphatic route or you have enterohepatic circulation. please see these two papers: "Oral particle uptake and organ targeting drives the activity of amphotericin B nanoparticles" and "Hemolytic and pharmacokinetic studies of liposomal and particulate amphotericin B formulations"
Author Response
Replies to Peer Reviewer: 2
(1) Have you considered to use Design of experiments instead of the OFTA (One factor at a time) methodology? I think this will increase the optimisation step of your formulations.
Answer: Thank you very much for your suggestion. We will do more consideration and improvement in the future studies.
(2) Why did you start the DSC at 50 degrees?
Answer: Thank you very much for your suggestion. Before the experiment, we know from some related references that the melting point of the paclitaxel is about 220 °C, so in order to save test time, we set the equilibrium and start temperature of DSC test as 50°C.
(3) The FTIR and Raman figures are not visible.
Answer: Thank you very much for your suggestion. I have uploaded FTIR and Raman figures again.
(4) You are claiming that you have a new polymorph of PTX based on the different Bragg peaks. Then, it is required to check the crystal structure of the new polymorph and uploaded into the Cambridge data base.
Answer: Thank you very much for your suggestion. At present, we are still lack of some data, unable to upload for the time being. But we will supplement the relevant data in the future.
(5) What is the peak at 221 degrees of your DSC? Are you sure the PTX does not degrade before melting?
Answer: Thank you very much for your suggestion. The DSC thermogram of the PTX revealed one endothermic peak at 221.99 °C and one exothermic peak at 244.6°C, which were attributed to the melting and decomposition of the PTX, respectively. Therefore, it can be explained that the PTX is melted first and then decomposed in the detection process.
(6) You should explain why you are getting your second peak in the PK study. I see two main reasons one that your drug is absorbed by lymphatic route or you have enterohepatic circulation. please see these two papers: "Oral particle uptake and organ targeting drives the activity of amphotericin B nanoparticles" and "Hemolytic and pharmacokinetic studies of liposomal and particulate amphotericin B formulations"
Answer: Thank you very much for your suggestion. the PTX crystal powders as an oral formulation had a second peak in the PK study, which might be due to the liver and intestine circulation or gastric emptying according to a certain time, so that the drug reached the small intestine twice, resulting in two times into the blood, which led to the appearance of double peaks. Relevant content has been added to the manuscript.
Reviewer 3 Report
The authors demonstrate interesting sets of data on the preparation of paclitaxel crystal by employing ionic liquids. The paper reports quite new approach to prepare new PTX crystal and its characteristics for improving oral bioavailability. Therefore, the paper is worthy of publication. However, it needs to be modified to address the following issues.
- The scientific background of the selection of 1-hexyl-3-methylimidazolium bromide (HMImBr) among ionic liquids should be stated in the introduction section.
- Did the authors consider the effect of particle sizes of PTX? What were the particle sizes of raw PTX and PTX crystal?
- How stable was PTX in HMImBr solution?
- How stable was PTX crystal during storage?
- The authors should measure the equilibrium solubilities of raw PTX and PTX crystal.
- It is well known that the oral bioavailability of paclitaxel is limited by not only the low -aqueous solubility but also p-glycoprotein. For this reason, improving oral bioavailability of paclitaxel should consider this.
- What was the reason of the second peak in plasma concentration of PTX at near 10 h?
- There are numerous reports on the improved solubility of PTX and its oral bioavailability. Thus, the authors should discuss their data by comparing to these reports.
- All figure legends should be more descriptive.
Author Response
Answer to reviewer 3
(1)The scientific background of the selection of 1-hexyl-3-methylimidazolium bromide (HMImBr) among ionic liquids should be stated in the introduction section.
A:Thank you very much for your suggestion. Related content has been added to the introduction section.
(2)Did the authors consider the effect of particle sizes of PTX? What were the particle sizes of raw PTX and PTX crystal?
A: Thank you very much for your suggestions. In this article, the particle size of PTX may have little effect on the improvement of its solubility and bioavailability, but the main reason is that the crystal structure of PTX crystalline powder has changed, and its crystallinity is significantly lower than PTX. Crude PTX (as shown by XRD) helps to improve its solubility and bioavailability. In addition, since the PTX crystal powder is crystalline rather than spherical, there is no way to determine its particle size. However, it can be seen from the SEM results that the particle size of the PTX crystal powder is smaller than that of the original PTX.
(3) How stable was PTX in HMImBr solution?
A:Thank you very much for your suggestion. First of all, the time for our preparation of PTX crystal powder is very short, so the liquid medicine will be used quickly. In addition, the remaining drug solution will not precipitate even if it is placed in an environment of 4°C for a long time, and can be detected by the HPLC system, so the stability of PTX in the HMImBr solution is very good.
(4)How stable was PTX crystal during storage?
ç”:Thank you very much for your suggestion. We will do some related research in the future.
(5)The authors should measure the equilibrium solubilities of raw PTX and PTX crystal.
A:Thank you very much for your suggestion. The equilibrium solubility of raw material PTX and PTX crystals in water is about 1.16μg/ml and 6.05μg/ml, respectively, and the equilibrium solubility in artificial gastric juice is about 10.42μg/ml and 24.65μg/ml, respectively.
(6)It is well known that the oral bioavailability of paclitaxel is limited by not only the low -aqueous solubility but also p-glycoprotein. For this reason, improving oral bioavailability of paclitaxel should consider this.
A:Thank you very much for your suggestion. We will do more considerations and improvements in future research.
(7)What was the reason of the second peak in plasma concentration of PTX at near 10 h?
A:Thank you very much for your suggestion. The PTX crystal powder in the oral preparation showed a second peak in the PK study, which may be caused by the liver and intestinal circulation or gastric emptying for a certain period of time. Therefore, the drug reached the small intestine twice, causing the blood to enter the blood twice, resulting Double peaks appear.
(8)There are numerous reports on the improved solubility of PTX and its oral bioavailability. Thus, the authors should discuss their data by comparing to these reports.
A:Thank you very much for your suggestion. Related content has been added to the introduction section.
(9)All figure legends should be more descriptive.
A:Thank you very much for your suggestion. I have made appropriate modifications and additions
Round 2
Reviewer 1 Report
I think the authors provided adequate answers to the comments raised by this reviewer and accordingly I recommend publication.
Reviewer 2 Report
Comments have been addressed correctly except the fact that the new crystal structure should be uploaded in the CSD database.